# Neurons in the bat auditory cortex encode class and complexity of future vocalizations
Susanne S. Babl [1,2,3] ✉, Dennis Röhrig[3] & Julio C. Hechavarría [1,2,3] ✉

Vocal production is a complex behavior across the animal kingdom that relies on coordinated motor and auditory networks. However, the contribution of sensory areas in vocal control remains poorly understood. Here, we investigated vocalization-related activity in a major auditory processing area in the mammalian brain, the auditory cortex. We recorded neuronal spike rates in an audio-vocal specialist, the bat *Carollia perspicillata*, which emits vocalizations to echolocate and for social communication. We found distinct spike patterns encoding echolocation and communication calls several hundred milliseconds before vocal onset. Neuronal activity not only contained information about the vocalization category, but also reflected temporal complexity, specifically the syllable amount in a communication sequence. These vocalization-specific firing patterns were dependent on the neurons' frequency receptive field. Our results indicate that single neurons in the auditory cortex not only process acoustic information but also encode the class and temporal structure of future vocal output.

Bats have long been investigated for their astounding ability to sense their environment through echolocation, a process in which they emit short vocalizations in the ultrasound range and gather information about their surroundings from the returning echoes. The characteristics of echolocation calls vary widely between species as they reflect adaptations and fine-tuning to specific habitats and environmental conditions. These auditory specialists produce, however, several other forms of vocalizations which, in contrast, serve the purpose of social communication. A bat's repertoire of communication calls can be highly diverse, with some species producing complex songs[1] and showing evidence of vocal learning[2,3]. The vocalizations of Seba's short-tailed bats (*Carollia perspicillata*), a frugivorous species inhabiting the neotropics, have been well studied in recent years[4–7]. The majority of their communication (COM) calls and their frequency-modulated echolocation (ECHO) pulses differ markedly in frequency composition, with ECHO pulses peaking above 60 kHz, and COM calls peaking between 10 and 40 kHz.

The brain circuitry and neuronal activity underlying vocal behavior in bats have been subject of many studies over the past years (reviewed by Babl et al.[8]). Yet to date, the neuronal mechanisms and network interactions that lead up to the production of ECHO or COM calls have not been fully understood. Several regions in the brainstem, such as the periaqueductal grey or the superior colliculus, have been identified to play a crucial role in controlling vocalizations[9–12]. But as vocal production and sensory processing are closely intertwined, especially in bats, regions along the auditory pathway are also likely to be involved in the control and fine-tuning of vocalizations. The auditory cortex (AC), the highest processing stage in the canonical auditory pathway, has been studied intensively in bats and other mammals in the context of vocal production or other self-produced sounds.

In primates and rodents, neuronal activity in the AC is typically modulated during vocal production. A recent study in mice could show that different populations of neurons either increase or decrease their firing rate already several hundreds of milliseconds before the onset of an ultrasound vocalization[13]. In monkeys, neurons that reduce their activity during vocalizations seem to be more predominant[14,15]. Such modulation is not limited to vocalizations but has also been observed during other self-generated, predictable sounds[16]. Compared to the response to external sounds, the activity to self-produced sounds can often be attenuated[17], which may be mediated by inhibitory efference copies originating in the motor system[18,19]. In addition, a recent study in marmoset monkeys indicated that activity in the AC can also play an active role in modulating vocal output by demonstrating that electrical stimulation in AC subregions can elicit pitch shifts during ongoing vocalization[20].

In bats, the AC presents several specializations to auditory processing that could play a further role in vocal production. Neurons in the AC are not

[1]AG Brain & Behavior, Institute of Biology, Freie Universität Berlin, Berlin, Germany. [2]Ernst-Strüngmann-Institute for Neuroscience, Frankfurt am Main, Germany. [3]Institute for Cell Biology and Neuroscience, Goethe University Frankfurt, Frankfurt am Main, Germany. ✉e-mail: s.babl@fu-berlin.de; j.hechavarria.cueria@fu-berlin.de

only sensitive to echo-delays, the time from the emission of an echolocation call until the echo return[21,22], but also track temporal structures of perceived vocalizations[23]. Temporal information encoded in the AC may be necessary to adjust vocal timing parameters. Accordingly, lesions in the AC have been shown to affect ECHO timing and rate[24]. Furthermore, AC neurons are spatially arranged according to their frequency-tuning profile, often with large areas dedicated to processing high-frequency sounds in the range of the bat's ECHO pulses[25,26]. These neurons are recruited during the perception of vocalizations displaying call-specific firing patterns depending on the frequency composition and context[23,27]. It is possible that these neurons also play a role in adapting and fine-tuning self-produced vocalizations. Indeed, recent work in *C. perspicillata* showed that AC local field potentials (LFP) differ in their oscillatory power already before vocal onset, depending on whether an ECHO or a COM call is about to be produced[28]. But in what way individual AC neurons may encode and control the production of different vocalizations is still unknown.

To address this question, we performed neuronal recordings in the AC of *C. perspicillata* while the bats were spontaneously producing ECHO and COM calls. We found that neuronal spike patterns preceding vocalizations already gave indication about the emitted call class and about their quantity (a single call vs. a sequence of pulses or syllables). Neurons' vocalization-specific firing before and after call onset was related to their individual frequency-tuning profiles. This indicates that the AC carries crucial information about vocalizations already before their production, with specific neuronal populations specializing in encoding different vocalization categories.

## Results
### Bats produce echolocation and communication calls as single vocalizations or in sequences

To study AC neuronal coding during vocal production, awake bats were placed in a custom-made holder for head-fixation, and their spontaneous vocalizations and neuronal activity were recorded inside an acoustically and electrically shielded chamber. We first analyzed the bats' vocalizations and categorized them into ECHO and COM according to their spectrogram properties (see "Methods"). ECHO pulses were easily identifiable by their two downward frequency-modulated harmonics with a peak frequency of ~80 kHz (Fig. 1A left). COM calls were more variable but typically had a peak frequency lower than 50 kHz (Fig. 1B). A typical COM syllable that bats emitted in our setup is shown in Fig. 1A (right). Frequency compositions of vocalizations were similar across individuals (Fig. S1A, C). We observed that bats often emitted either a single (SIN) call or a short sequence (SEQ) containing multiple syllables or pulses. We therefore categorized calls further into (1) SIN ECHO, (2) SIN COM, (3) SEQ ECHO and (4) SEQ COM (see Methods, Fig. 1D-G). A SEQ was characterized by at least three COM syllables or ECHO pulses, respectively, within a period of 500 ms. Typically, COM syllables within a SEQ were produced at a faster rate with a shorter interval than ECHO pulses (Fig. 1I, $p < 0.001$, Wilcoxon rank-sum test). To analyze neuronal activity in one second around each call category, we restricted our analysis to vocalizations with a silent 500 ms period (no vocalization, no noise) before their onset. Vocalizations followed by noise or a different call class than the first in 500 ms after onset were equally excluded (see "Methods"). According to these criteria, 631 vocalizations from these four categories were identified across all sessions ($n = 32$) and animals ($n = 10$), specifically 196 SIN ECHO, 244 SIN COM, 33 SEQ ECHO and 158 SEQ COM (Fig. 1J). Male and female bats contributed a comparable number of vocalizations to the dataset (Fig. S1E).

### Neurons in the AC code for upcoming vocalizations

To investigate the neuronal activity in the AC underlying different vocalization categories, we next analyzed firing rates of spike sorted neurons (see "Methods") in a period of 500 ms before (pre) until 500 ms after (post) each of these vocalization onsets. First, we compared firing rates of individual neurons recorded during multiple call events of SIN ECHO and SIN COM. In many neurons, we observed distinct firing patterns depending on the

vocalization category. These differences were evident already several hundred milliseconds before call onset (Figs. 2A, S2A, B, G).

To directly compare these firing rates, we computed a modulation index for every neuron in pre and post time windows (see Methods). A positive modulation index indicates a higher firing rate for SIN ECHO, whereas a negative modulation index indicates a higher firing rate for SIN COM. A value of 0.3 (or −0.3, respectively) would indicate a doubling in firing rate for one call category compared to the other. We used this threshold to identify neurons with a clear firing increase for one category. With this method, we found that 25% of neurons (42 out of 166) fired stronger before a SIN ECHO compared to SIN COM (Fig. 2B). Another population of neurons fired more strongly before SIN COM compared to SIN ECHO, with 20% (34/166) showing a clear increase. After vocalization onset, 25% of neurons (42/166) fired stronger for SIN ECHO vs. SIN COM, whereas 15% (25/166) fired more for SIN COM.

When we averaged the activity across neurons, we observed overall higher firing rates for SIN ECHO than for SIN COM (Fig. 2C). A two-way ANOVA (call category x time) using firing rates in 50 ms time bins revealed a significant effect of category ($p < 0.0001$), a significant effect of time ($p < 0.05$) and a significant interaction ($p < 0.001$). Statistical comparisons for individual time bins showed the clearest differences at 325 to 275 ms before call onset and at 325 to 375 ms after call onset ($p < 0.01$, Wilcoxon signed-rank test). To assess whether vocalization-related responses showed overall excitation or suppression following vocal onset, we computed a modulation index relative to a baseline period preceding the pre-vocal window. While both suppressed and excitatory responses were observed, the distribution of modulation indices was skewed toward positive values, indicating a predominance of increased firing rates following the onset of vocal production (Fig. S2K).

To examine vocalization-specific population-level structure in more detail, we performed principal component (PC) analysis on these neurons' firing rates around vocalization onset (see "Methods"). The first 3 PCs, which together explained approximately 40% of the variance, revealed the dominant underlying vocal activity patterns and demonstrated a clear separation of the trajectories for SIN ECHO and SIN COM already before call onset (Figs. 2D, S3A). To assess the contribution of specific neuron groups to these population trajectories, we removed neuronal subsets from the analysis (see "Methods"). First, we removed 30% of neurons with the strongest modulation indices for either call category (high discriminators). This markedly reduced the difference between trajectories, especially for the first PC. In contrast, removing 30% of neurons with the weakest modulation indices (low discriminators) did not produce any clear change in population trajectories. To assess the difference in magnitude between SIN ECHO and SIN COM trajectories within PC space, we computed the Euclidean distance ($d$) for the first 3 PCs (see "Methods"). This revealed the largest difference for the complete dataset with all neurons ($d = 2.47$), a slight reduction when low discriminator neurons were removed from the dataset ($d = 2.28$) and a strongly reduced difference when high discriminator neurons were removed ($d = 1.45$; Fig. S3E). Additionally, we used the full recorded neuronal population to compute support vector machine (SVM) models and predict call category (see "Methods", Fig. S3I). These models were able to significantly predict produced vocalizations pre and post vocal onset. Taken together, this indicates that neuronal subpopulations in the AC clearly differentiate between ECHO and COM calls already before their production.

Next, we asked whether the quantity of produced syllables in a call is reflected in neuronal activity. To this end, we first compared firing rates of neurons for SIN COM vs. SEQ COM. Also in this case, we observed distinct firing patterns in several neurons depending on the call category, which was already evident before vocalization onset (Figs. 2E, S2C, D, H). When we computed modulation indices for pre and post time periods, as described above, we found that 18% of neurons (19 out of 108) increased their firing rate before a SEQ COM compared to SIN COM. On the other end, 12% of neurons (13/108) fired more strongly before a SIN COM compared to SEQ COM. After vocalization onset, 31% of neurons fired

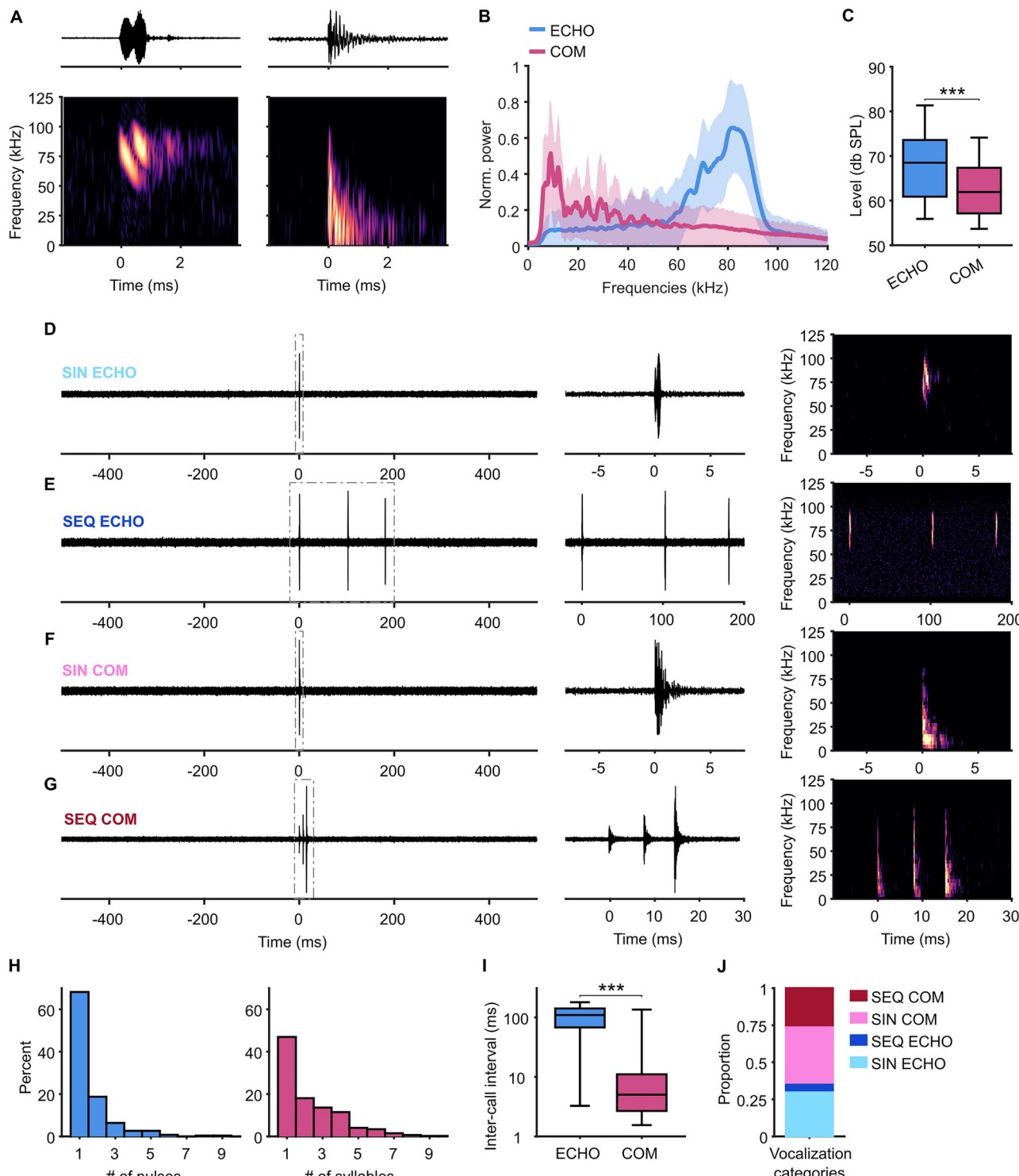

**Fig. 1 | ECHO and COM calls are produced in SIN and SEQ configurations.**
**A** Oscillogram (top) and spectrogram (bottom) of an example ECHO (left) and COM (right) call. **B** Average power spectra of recorded ECHO and COM vocalizations showing mean (line) and standard deviation (shaded area). **C** Level of ECHO and COM. Box plots show median (line), 25th and 75th percentile (box) and whiskers to the minimum and maximum values within 1.5 times the interquartile range. **D–G** Examples of the 4 call categories SIN ECHO (**D**), SEQ ECHO (**E**), SIN COM (**F**) and SEQ COM (**G**) with oscillograms of the full selected time period from 500 ms before until 500 ms after call onset (left). Note that calls in these categories are always preceded by 500 ms of silence. SIN vocalizations consist of only a single pulse or syllable, SEQ consist of at least 3 pulses or syllables within 500 ms. Dashed squares indicate period for the zoomed in oscillogram (middle) and spectrogram (right). **H** Percentages of vocalizations after a 0.5 s silent period containing a certain number of ECHO pulses (left) or COM syllables (right). **I** Inter-call interval is much longer in ECHO compared to COM vocalizations, ***$p < 0.001$ Wilcoxon rank-sum test. **J** Proportions of recorded vocalizations that fall into the 4 described categories.

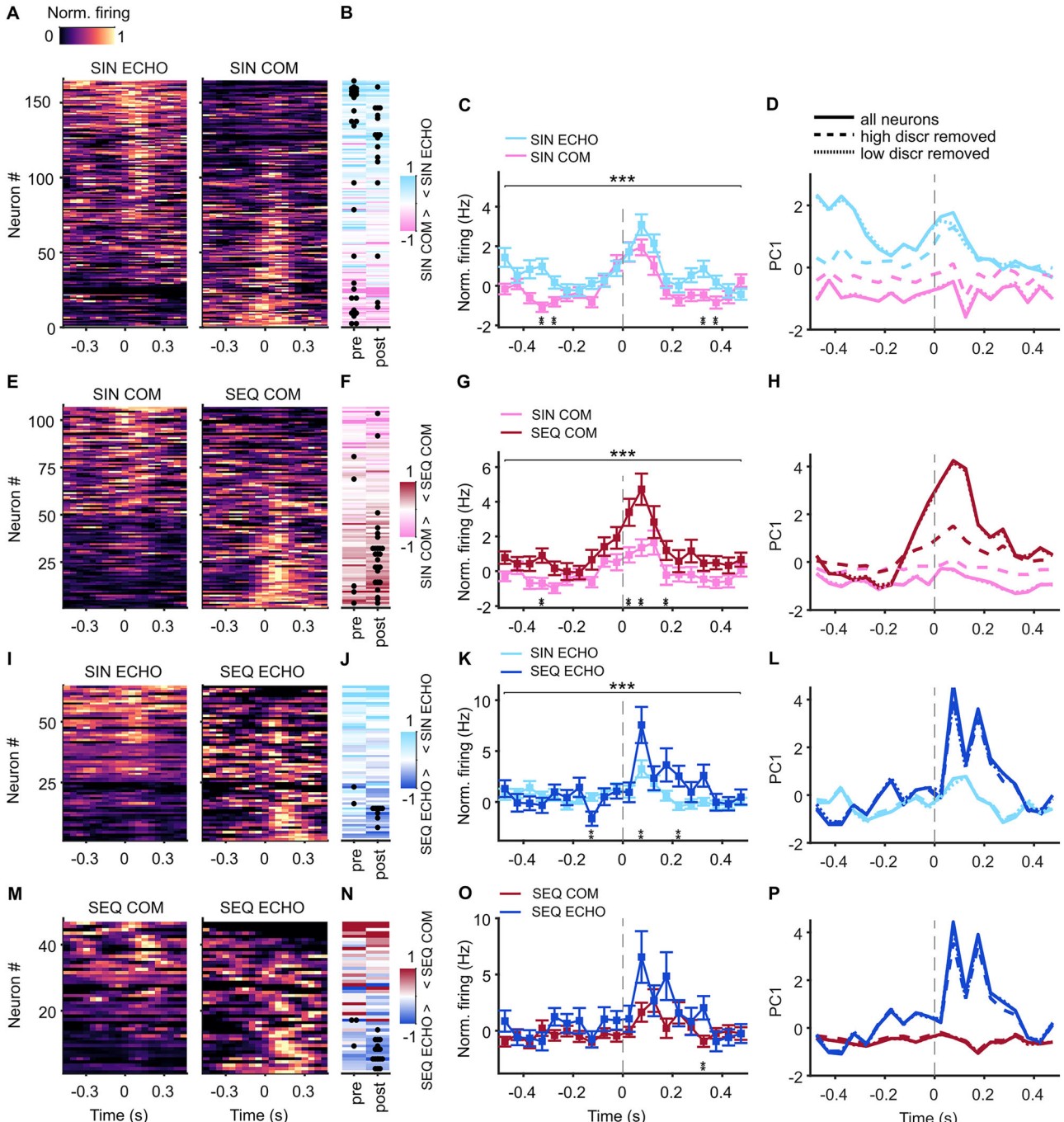

**Fig. 2 | AC neurons show vocalization-specific firing patterns starting before their onset. A** Color plots showing normalized firing rates of neurons recorded during call events of SIN ECHO (left) and SIN COM (right). Rows of left and right plot show activity of the same neuron around the two call categories. Neurons are sorted to the largest average difference in firing between the two call categories. **B** Modulation indices for averaged firing rates pre and post call onset indicating firing differences between call categories. Rows show neurons as in (**A**). Positive values (blue) indicate higher firing rates for SIN ECHO, negative values (pink) indicate higher firing rates for SIN COM. Black dots indicate significant modulation with *$p < 0.05$, **$p < 0.01$, ***$p < 0.001$, Wilcoxon rank-sum test. **C** Averaged firing rates ±SEM of neurons shown in (**A**) normalized by subtracting firing in the baseline period 1-0.5 s before call onset. Top asterisks indicate significant effect of call category (***$p < 0.001$, two-way ANOVA, see text), bottom asterisks indicate significant difference in the respective time bin (**$p < 0.01$, Wilcoxon signed-rank test). **D** 1st PC scores over time from PC analysis on firing rates to call categories of neurons in (**A**). Solid lines indicate dataset with all neurons, dashed lines indicate dataset without high discriminator neurons (30% of neurons with highest modulation indices), dotted lines indicate dataset without low discriminator neurons (30% of neurons with lowest modulation indices). **E–H** Figures as in (**A–D**) for neurons recorded during SIN COM and SEQ COM. **I–L** Figures as in (**A–D**) for neurons recorded during SIN ECHO and SEQ ECHO. **M–P** Figures as in (**A–D**) for neurons recorded during SEQ COM and SEQ ECHO.

more for SEQ COM, whereas 14% increased their firing rates for SIN COM (Fig. 2F).

When we analyzed firing rates across neurons, we found that average activity was higher for SEQ COM compared to SIN COM calls (Fig. 2G). A

two-way ANOVA (call category x time) showed a significant effect of category ($p < 0.0001$), a significant effect of time ($p < 0.001$) and a significant interaction ($p < 0.0001$). A comparison of individual time bins showed the most pronounced difference was at 25 ms to 75 ms post onset of the first call,

but also at 325 ms pre call onset ($p < 0.01$, Wilcoxon signed-rank test). PC analysis revealed clearly separated trajectories for SIN COM vs. SEQ COM for PC1-3, which diverged already bevor call onset (Figs. 2H, S3B; $d = 2.65$). When we removed high discriminator neurons, as described above, trajectories of call categories became much more similar ($d = 1.24$), whereas removing low discriminator neurons had little effect ($d = 2.57$; Fig. S3F). This suggests that neuronal activity in the AC differentiates between COM calls with one syllable and with multiple syllables already before the call is produced.

We then analyzed the firing rate of neurons recorded during SIN ECHO vs. SEQ ECHO. While we observed neurons with distinct firing rates for the two call categories (Figs. 2I, S2E, F, I), changes over time appeared more complex regarding the pre- and post-vocal time window. Before vocal onset, 25% neurons (16/65) increased their firing rates for SIN ECHO compared to SEQ ECHO, while 14% (9/65) showed stronger firing rates for SEQ ECHO than SIN ECHO. After vocal onset, these ratios reversed, and 29% (19/65) fired higher for SEQ ECHO, whereas only 20% (13/65) showed increased firing for SIN ECHO (Fig. 2J). This trend was also confirmed when we analyzed firing rates across all neurons (Fig. 2K). On average, firing rates pre vocal onset were slightly lower for SEQ ECHO compared to SIN ECHO, especially at 125 ms pre ($p < 0.01$, Wilcoxon signed-rank test). However, post onset, average firing rates were much higher for SEQ ECHO than for SIN ECHO. The most significant differences were observed at 75 ms and at 225 ms post vocal onset ($p < 0.01$). A two-way ANOVA (call category x time) showed a significant effect of category ($p < 0.001$), a significant effect of time ($p < 0.001$) and a significant interaction ($p < 0.01$). SVM population decoding of call category was already significantly above chance before vocal onset, but with lower accuracy than following vocal onset (Fig. S3K). PC analysis revealed the clearest trajectory differences after vocal onset (Figs. 2L, S3C). Interestingly, removing high discriminator neurons had little effect and was similar to removing low discriminators ($d$ full dataset $= 1.94$, $d$ with high discriminators removed $= 1.74$, $d$ with low discriminators removed $= 1.62$; Fig. S3G). This points towards complex coding patterns for SIN and SEQ ECHO, that are distinct from COM coding.

Finally, we analyzed neurons recorded during SEQ COM vs. SEQ ECHO (Fig. 2M). Modulations before vocal onset were evenly distributed, with 25% (12/48) firing more strongly for SEQ ECHO and 25% for SEQ COM. Post vocal onset, more neurons increased their firing for SEQ ECHO (38%, 18/48), and a smaller proportion increased firing for SEQ COM (25%, Fig. 2N). Comparing vocalization-related firing rates across neurons (Fig. 2O) did not reach statistical significance (two-way ANOVA (call category x time), no effect of category, $p = 0.98$, significant effect of time, $p < 0.05$, no interaction, $p = 0.43$). However, comparison for individual time bins showed some significant differences after vocal onset, especially at 325 ms ($p < 0.01$, Wilcoxon signed-rank test). PC analysis also revealed the clearest trajectory differences after vocal onset (Figs. 2P, S3D). Removing high or low discriminator neurons had similar effects ($d$ full dataset $= 2.21$, $d$ with high discriminators removed $= 1.97$, $d$ with low discriminators removed $= 1.92$; Fig. S3H).

In summary, these results suggest that neuronal spike rates in the bat AC encode different categories of vocalizations already several hundred milliseconds before the animal starts to vocalize. Neurons seem to differentiate not only between broad vocalization classes (ECHO and COM), but also between the production of single and multiple syllables or pulses.

## AC spike rates depend on syllable or pulse quantity

Especially for COM calls, firing rates seemed to overall increase when a SEQ compared to a SIN call was emitted. We therefore asked whether neural activity scaled with the number of syllables produced. To this end, we first identified call events depending on the number of COM syllables emitted within 500 ms and then computed spike rates of neurons that were recorded during several of such COM call events (see "Methods"). This revealed that several neurons increased their spike rate when higher numbers of syllables were emitted, often already in the time period before vocal onset (Fig. 3A, B).

We correlated the spike rates of each neuron pre and post each call event with the number of emitted syllables and found that on average, neurons showed a positive correlation between spike rate and syllable count. This effect was already visible before call onset and became more pronounced after call onset (Fig. 3C).

To analyze population activity in relation to the number of COM syllables, we then computed the spiking activity of every neuron recorded during every vocalization event in pre and post time windows. By using the spiking activity in all instances, we could include also neurons that were recorded only during very few call events (see "Methods"). When we analyzed neuronal firing in the pre-vocal window as a function of produced COM syllables, we found a gradual increase in activity with increasing syllable numbers (Fig. 3D left). Fitting a linear regression model confirmed this relationship, revealing a rising slope of neuronal firing across syllable counts (slope = 0.016, 95% CI = [0.006, 0.026], $p < 0.01$). When we analyzed neuronal firing to the same vocalization events after onset of the first call, the observed increase was even stronger (Fig. 3D right). Fitting a linear regression model showed a significantly rising slope (slope = 0.039, 95% CI = [0.026, 0.052], $p < 0.0001$).

We then asked whether a similar relationship existed between neuronal activity and emitted ECHO pulses. First, we looked for individual neurons that were recorded during a certain number of call events with various numbers of pulses. When we correlated the spike rates of neurons before call onset with the number of emitted pulses, we found however no significant relationship across the population. In contrast, after call onset, neural firing rates significantly correlated with pulse number (Fig. 3E, F, G). We then analyzed spiking activity of all neurons to all ECHO events and again found no clear increase or decrease in the pre-vocal window (Fig. 3H left). A linear regression model revealed no significant relationship to ECHO pulse counts ($p = 0.58$). Spiking in the post-vocal window, however, showed a significantly rising slope with the number of emitted pulses (Fig. 3H right; slope=0.11, 95% CI = [0.08, 0.13], $p < 0.0001$).

Taken together, this indicates that AC neuronal firing increases with the number of emitted COM syllables and ECHO pulses. In the case of COM calls, this relationship emerges already in the pre-vocal period, before any of the syllables are produced.

## Call-specific firing depends on a neuron's frequency tuning profile

We next wondered whether the call-specific firing patterns observed in individual neurons were related to other activity properties. Neurons in the AC are often specifically tuned to a certain frequency range, showing a clear firing preference for sound at their best frequency (BF). To get each neuron's frequency tuning profile, we played short pure tones ranging from 10 to 90 kHz at a fixed level (65 dB SPL) while recording neuronal activity (see "Methods"). We found that neurons showed a preference for low or high frequencies, with very few neurons responding best to intermediate frequencies (Fig. 4G-H). This is largely in accordance with the hearing threshold for *C. perspicillata*[29].

We then classified neurons in frequency tuning subgroups according to their response to the presented pure tones (see "Methods") and found that 32% of the recorded neurons (123/384) were high frequency tuned (HF) neurons, i.e. their BF was at least 50 kHz (Fig. 4A-B, I). 21% (79/384) were low frequency tuned (LF) with their BF below 50 kHz (Fig. 4C-D, I). The majority, however, showed firing peaks in the low and high frequency range (see "Methods"), and they were therefore called multipeak (MP) neurons (47% or 182/384; Fig. 4E-F, I), as described in previous studies[27,30]. This neuron type has been associated with non-tonotopically arranged high-frequency areas in the AC[21], which corresponds to our main targeted area. HF, LF and MP neurons also differed significantly in their bandwidth, with HF neurons covering the lowest octave space and the highest Q-factor, indicative of sharp tuning properties (Fig. S4A-D).

To test whether frequency tuning profiles of neurons were connected to their vocal activity patterns, we analyzed firing rates around call onset separately for neurons in HF, LF and MP subgroups. First, we used neurons

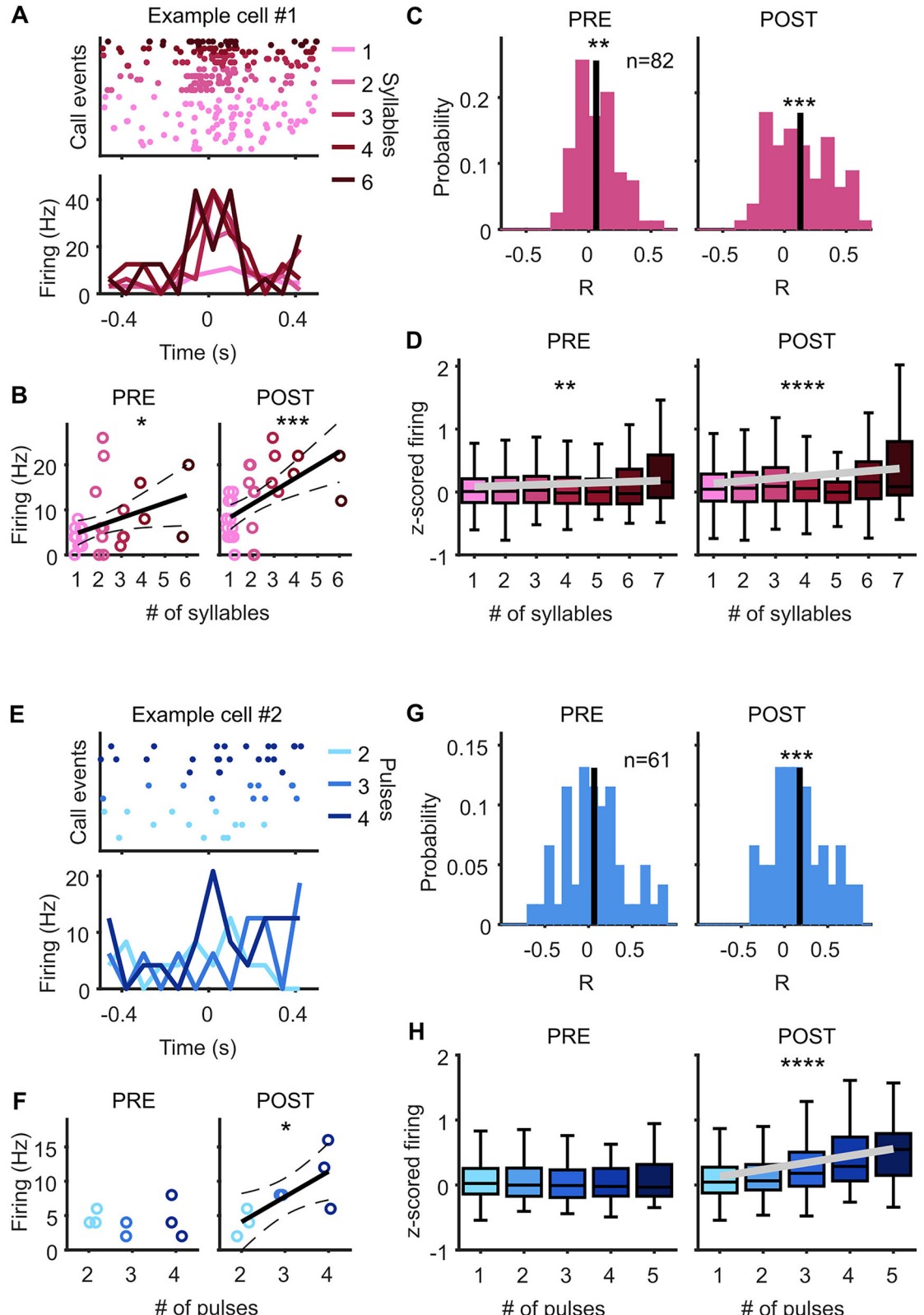

recorded during SIN ECHO and SIN COM and computed their firing modulation indices over time, indicative of the change in firing rate depending on the call category (see Methods). We noticed that LF neurons tended to fire more strongly when a SIN COM was about to be produced (Figs. 5A top, 5B). In contrast, HF and MP neurons fired more strongly around the onset of a SIN ECHO (Fig. 5A middle, bottom, 5B). To quantify

this effect, we computed time-averaged modulation indices for the pre- and post-vocal window and compared them across frequency groups (Fig. 5C). We found that both HF and MP neurons were significantly more active for SIN ECHO compared to SIN COM than LF neurons, which in contrast, were more active to SIN COM. These activity patterns were visible both in the pre- and post-vocal window. We then analyzed the modulation indices

**Fig. 3 | Neuronal activity increases with the number of emitted syllables or pulses.**
**A** Raster plot (top) and PSTH (bottom) of example neuron recorded during multiple COM call events with various syllable numbers. Firing rates around vocal onset are higher for higher syllable counts. **B** Firing rate of neuron shown in (**A**) significantly increases with the number of syllables, both in the pre-vocal (left) and post-vocal (right) window. Each dot indicates the firing rate to one call event. Solid lines indicate linear regression fit, dashed lines indicate 95% CI, asterisks indicate significance level of linear regression model with *$p < 0.05$ and ***$p < 0.001$. **C** Histograms showing Pearson's correlation coefficients of neurons recorded during multiple COM call events with different syllable numbers. On average, neurons show a positive

correlation in the pre-vocal (left) and post-vocal window (right). **$p < 0.01$, ***$p < 0.001$, Wilcoxon signed-rank test. **D** Population activity of call instances (neurons x call events) as a function of emitted syllables shows a positive relationship in the pre-vocal (left) and post-vocal (right) window. Black lines indicate linear regression fit, asterisks indicate significance level of linear regression model, **$p < 0.01$, ****$p < 0.0001$. Box plots represent median (line), 25th and 75th percentile (box) and whiskers extend to the minimum and maximum values within 1.5 times the interquartile range. **E–H** Figures as in (**A–D**) for ECHO call events. Firing rates increase with the number of ECHO pulses only after, but not before, call onset.

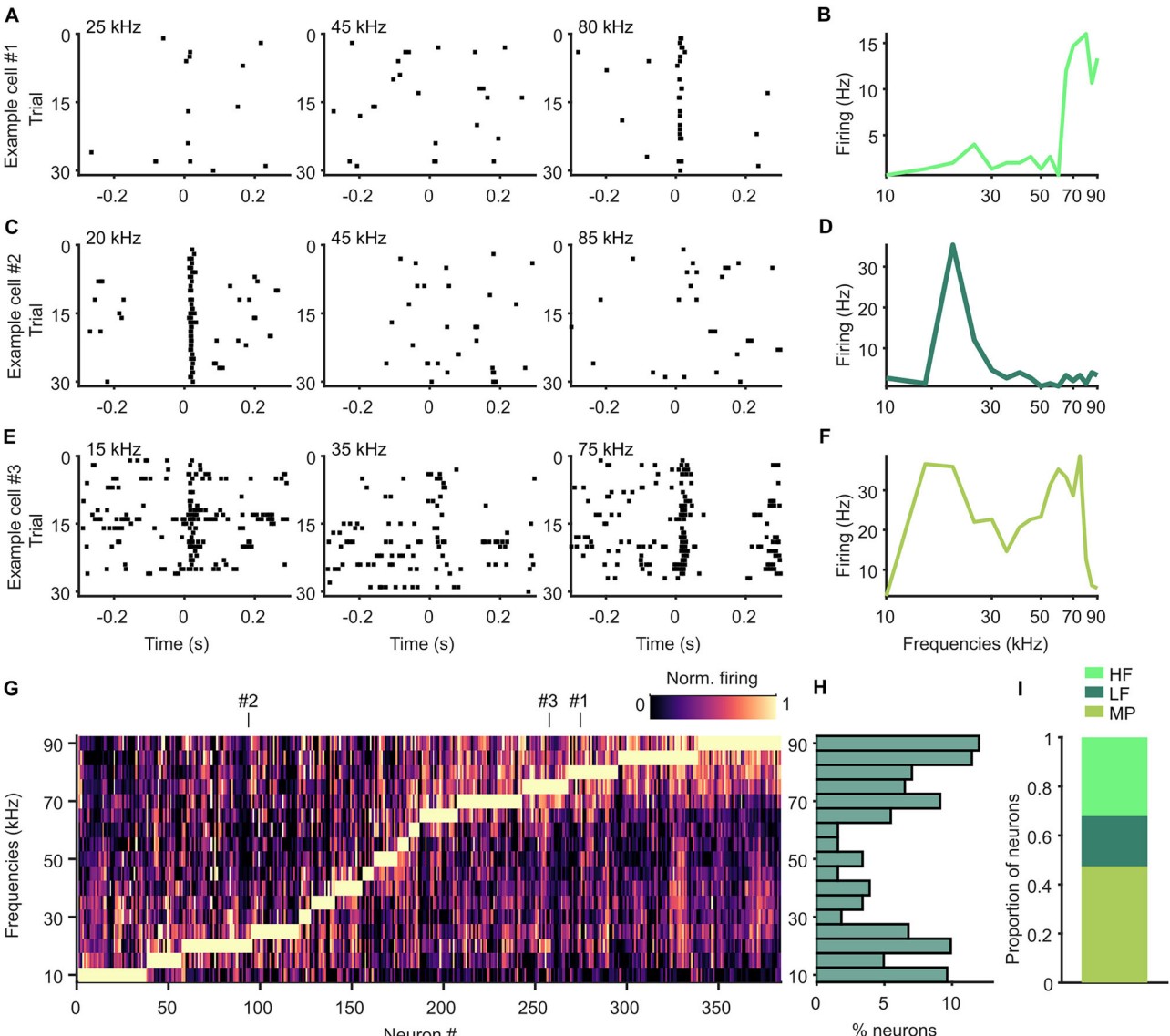

**Fig. 4 | Frequency tuning of neurons in AC. A** Raster plots of example cell showing spiking activity to presented pure tones with low (left), intermediate (middle) and high (right) frequency. Exact tone frequency is marked on top. **B** Frequency tuning curve of example neuron in (**A**) showing average firing rate to all presented pure tones. This neuron responds best to high frequency tones ( = HF neuron).

**C–F** Figures as in (**A, B**) but with LF example neuron (**C, D**) and MP example neuron (**E, F**). **G** Color plot showing tuning curves of all recorded neurons. Neurons are sorted according to their BF. Example cells from (**A–F**) are marked on top.
**H** Histogram showing percentage of neurons tuned to respective pure tone frequencies. **I** Proportions of HF, LF and MP neurons.

of neurons as a function of their BF and found a significantly increasing relationship (Fig. 5D). Neurons tuned to lower frequencies tended to be more active before and after production of a SIN COM, whereas neurons tuned to higher frequencies tended to be more active before and after the production of a SIN ECHO. Fitting a linear regression model confirmed this relationship, revealing a rising slope of call-dependent firing modulation across BF (Pre: slope = 0.077, 95% CI = [0.006, 0.148], $p < 0.05$; Post:

slope=0.068, 95% CI = [0.004, 0.131], $p < 0.05$). The tuning BW or the Q-factor did not show any relationship to the neurons' firing modulation (Fig. S4E, H).

These results align well with the spectral properties of the vocalizations, as COM calls typically contain lower frequencies than ECHO pulses. But is the frequency tuning profile of a neuron also informative of its coding for different syllable quantities of the same call class? To address this question,

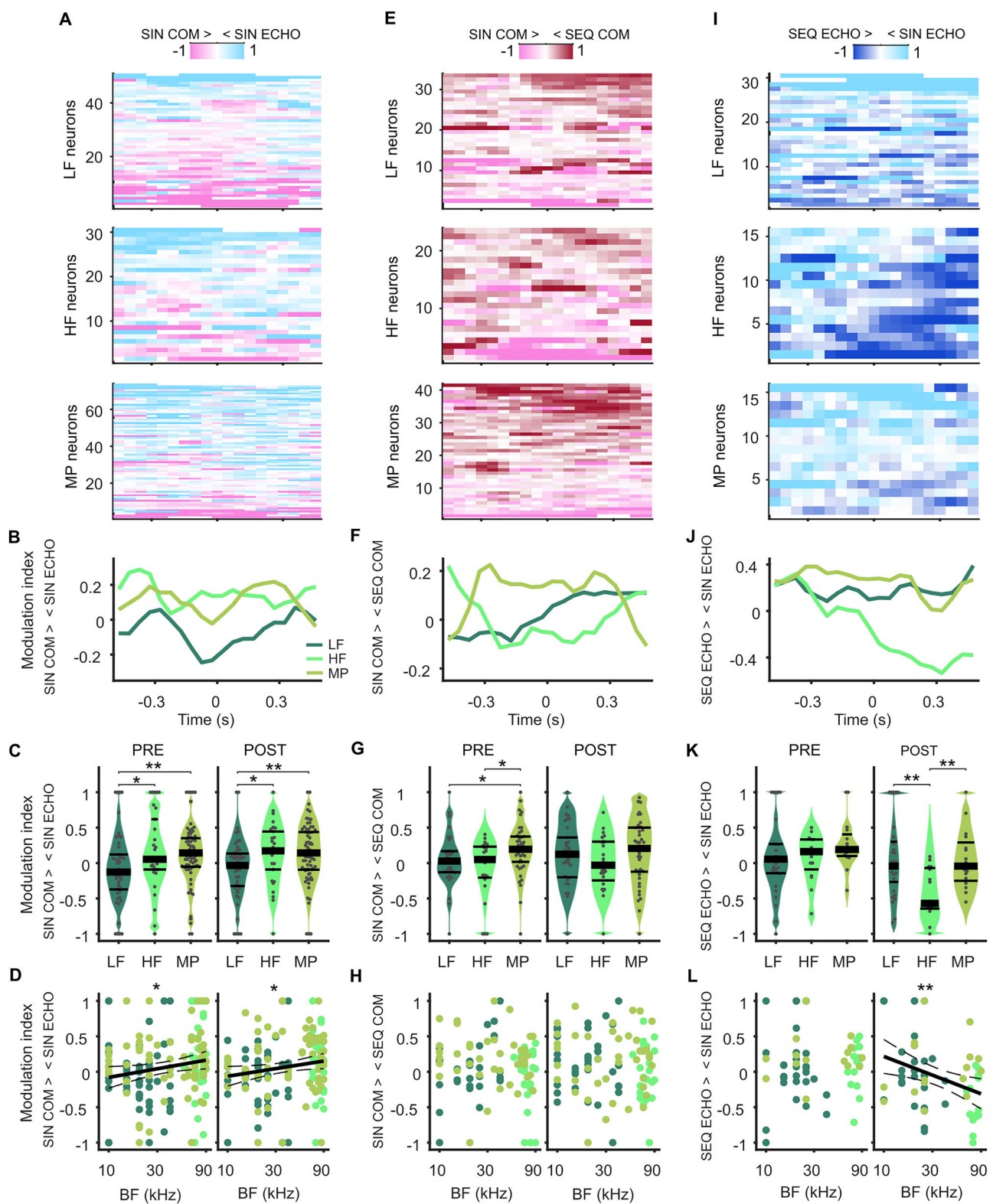

we analyzed neurons recorded during SIN COM and SEQ COM according to their frequency subgroups (Fig. 5E). We observed that while LF and HF neurons could be active to either SIN COM or SEQ COM, many MP neurons fired more strongly when a SEQ COM was about to be produced. In the pre-vocal window, MP neurons were significantly more modulated in the direction of SEQ COM than LF and HF neurons (Fig. 5F, G). While a similar trend was observed in the post-vocal window, it did not reach

statistical significance ($p > $ =0.15, Wilcoxon rank-sum test). In accordance with these results, the BF of the neurons did not change significantly with their firing modulation for SIN or SEQ COM (Fig. 5H; $p > 0.42$, linear regression model). But the tuning BW increased significantly with the firing modulation in the post-vocal window (Fig. S4F).

We then asked whether frequency tuning played a role in coding for ECHO pulse quantities and analyzed neurons recorded during SIN ECHO

**Fig. 5 | Vocal firing patterns depend on a neuron's frequency tuning. A** Call-type-dependent firing modulation in neurons recorded during SIN COM and SIN ECHO over time. Neurons are categorized as LF (top), HF (middle) and MP (bottom) according to their frequency tuning properties and sorted according to the largest firing differences in call categories. Positive values (light blue) indicate higher firing rates for SIN ECHO, negative values (pink) indicate higher firing rates for SIN COM. **B** Modulation indices over time averaged across LF, HF and MP neurons. **C** Modulation indices for averaged firing rates pre (left) and post (right) call onset indicating firing differences between call categories for neuron frequency subtypes. MP and HF neurons fire more strongly to SIN ECHO compared to SIN COM than LF neurons. Plots show density, modulation indices of individual neurons (grey dots), median (thick black lines), 25th and 75th percentiles (thin black lines). *$p < 0.05$, **$p < 0.01$, Wilcoxon rank-sum test. **D** Call-type-dependent firing

modulation changes significantly depending on the BF pre (left) and post (right) vocal onset. Each dot indicates the modulation index and the BF of a neuron. Solid lines indicate linear regression fit, dashed lines indicate 95% CI, asterisks indicate significance level of linear regression model with *$p < 0.05$. **E–H** Figures as in (**A–D**) but for neurons recorded during SIN COM and SEQ COM. Positive values (dark red) indicate higher firing rates for SEQ COM, negative values (pink) indicate higher firing rates for SIN COM. MP neurons are more active before SEQ COM compared to SIN COM than LF or HF neurons. **I–L** Figures as in (**A–D**) but for neurons recorded during SIN ECHO and SEQ ECHO. Positive values (light blue) indicate higher firing rates for SIN ECHO, negative values (dark blue) indicate higher firing rates for SEQ ECHO. HF neurons are more active after onset of SEQ ECHO than SIN ECHO than LF or MP neurons.

and SEQ ECHO according to their frequency groups. While before vocal onset, neuron subgroups showed mixed activity patterns, we observed a strong firing modulation post-vocally specifically in HF neurons (Fig. 5I, J). Almost all of these neurons showed stronger firing rates after the onset of SEQ ECHO compared to SIN ECHO. Modulation indices in HF neurons were significantly different from LF and MP neurons, but only in the post-vocal window (Fig. 5K). Accordingly, firing modulation for SIN ECHO became significantly lower with increasing BF (Fig. 5L; slope = −0.167, 95% CI = [−0.286 −0.047], $p < 0.01$, linear regression model), with decreasing BW, and with increasing Q-factor (Fig. S4G, J). In the pre-vocal window, we could not find a significant relationship between these parameters ($p > 0.54$).

Taken together, the results described above indicate that vocalization-specific firing depends on a neuron's frequency tuning profile. This relationship can already emerge before vocal onset and encode differences between call class as well as syllable quantity.

## Discussion

In this study, we examined neuronal activity in the bat AC during spontaneous vocal production. We found that individual AC neurons encode information about upcoming vocalizations several hundred milliseconds before they are emitted. Spike patterns differed not only between broad vocalization classes (ECHO vs. COM) but also reflected the number of syllables to be produced. These vocalization-specific activity patterns depended on a neuron's frequency tuning profile. Specifically, neurons with a higher BF were more likely to increase firing before an ECHO call. Broadly tuned MP neurons were modulated depending on whether a single-syllable or multi-syllabic COM call was about to be produced.

Previous studies have already identified a connection between AC activity and sound production. In vocalizing mice, it was shown that a large number of neurons in the AC starts to modulate firing rates approximately half a second before vocal onset. This was even the case when mice were deafened and could not hear their own vocalizations[13]. Similarly, in marmoset monkeys changes in AC spiking and local field potentials were observed as early as one second before the animal started to vocalize[14,15,31]. When the monkeys shifted the pitch during an ongoing vocalization to compensate for frequency-modulated playback of their own calls, several AC neurons were modulated depending on the strength of vocal compensation. A vocal pitch shift could also be elicited by electrical micro-stimulation in multiple AC subregions[20]. Together, this indicates that the AC exhibits pre-vocal activity that reflects information about the upcoming vocal output. Our findings confirm and extend previous results across taxa by demonstrating that the AC encodes not only frequency content but also temporal information about future vocalizations. Importantly, the pre-vocal firing pattern observed here cannot be explained by a uniform motor-related modulation as it carries information about upcoming call type and temporal structure. This suggests that AC activity reflects specific aspects of planned vocal output rather than general gain modulation of sensory responses.

The observed preparatory signals could be transmitted from motor areas, for example, in the frontal cortex, where call-type specific activity on the level of local field potentials has been found in previous studies from our

group[28,32]. There is evidence in multiple species for direct projections from frontal cortical regions to the AC[33–35], and simultaneous recordings of local field potentials in the frontal cortex and AC revealed that specifically in the pre-vocal period, information flow from frontal areas predominates[28].

Beyond the cortex, subcortical circuits have been described to play a crucial role in controlling vocal output. The periaqueductal gray is known to be critical for vocal production across mammals[36,37] and potentially controls different types of vocalizations through distinct neuronal populations[9–11]. Furthermore, the bat superior colliculus has been shown to encode specific properties of upcoming ECHO pulses[38,39] and play a crucial role in their production[10,12]. Apart from the midbrain, also neuronal activity in the bat cerebellum contains information about the upcoming vocalization type[40]. These findings raise the possibility that subcortical networks contribute to pre-vocal modulation in the AC, though the underlying circuitry remains to be determined.

The observed pre-vocal activity in the AC could play a role in priming the network for processing the upcoming vocalization and its associated sensory feedback. While all animals experience feedback from their environment to a produced sound, this is elevated to a matter of central importance for bats as they use returning echoes following their ECHO vocalizations to navigate. Our results demonstrate clear differences in firing patterns preceding ECHO and COM calls, which could be reflective of the distinct behavioral and sensory demands of these two vocalization modes. Echoes return only a few milliseconds after call emission and require a cascade of fast processing throughout the auditory pathway to extract relevant information about the environment. We found that on average, firing rates were higher before and following an ECHO compared to a COM call. This could reflect an active state of these neurons in anticipation of processing the returning echoes. Firing rate enhancement during echoes following vocalization has been shown before, for example in the inferior colliculus of horseshoe bats[41]. This is also compatible with our observation that predominantly neurons tuned to high frequencies fired more strongly before and after an ECHO pulse, as these neurons are likely to be involved in echo processing due to their high responsivity in the range of echo frequencies. It remains, however, difficult to fully disentangle whether the observed firing differences between ECHO and COM calls reflect mainly acoustic features, call category, or an interaction of both since these call classes differ both in behavioral context and in their spectral content.

As in echolocating species, vocal production is tightly coupled to navigation, the observed pre-vocal firing patterns may be embedded within a broader framework predicting audiospatial information. Recent findings show that hippocampal spatial and goal coding is informed by ECHO calls and encodes future spatial trajectories during flight[42–44]. It is possible that predictive coding in the AC and hippocampus reflects coordinated planning of sensory sampling and movement, particularly during echolocation. Within this framework, the pre-vocal activity observed here may form part of a more general mechanism that supports rapid sensorimotor integration.

It should be noted that the vocal repertoire of *C. perspicillata* is very diverse and strongly dependent on behavioral context[4–6]. When the bats are recorded in our setup, where they are restrained in movement and without

contact to conspecifics, they typically emit COM calls as presented in Fig. 1 (for further examples see refs. 32,40). While these short, low-frequency COM calls have been observed in freely moving animals, their ethological role remains unclear. In more natural conditions, particularly during social interaction or flight, the repertoire is expected to be broader, including a wider range of COM types and more frequent SEQ ECHO.

Our data suggest that firing differences between SIN ECHO and SEQ ECHO emerge mostly after the first pulse is emitted, while in contrast, changes between SIN COM and SEQ COM were clearly apparent in the pre-vocal window. These observations may indicate that these vocalizations are encoded at different timescales: As inter-call intervals within SEQ ECHO are typically longer than within SEQ COM, AC neurons could encode each ECHO pulse only shortly before its production, whereas rapidly produced COM syllables may be represented more as one sequence-level event. In freely moving or flying animals, where SEQ ECHO are very abundant, this could be investigated more in detail, as the increased number of sequences could reveal clearer temporal dynamics of spike patterns. Overall, we expect that the identified coding principles extend to natural conditions, but within a richer and more dynamic behavioral and acoustic context.

In addition, we also observed modulation of neuronal activity within vocalization categories depending specifically on the number of produced syllables or pulses. Previous studies in the Big brown bat (*Eptesicus fuscus*) had shown that spike patterns in the superior colliculus differ depending on temporal properties of upcoming ECHO pulses[38,39]. Our results reveal that in the AC, pre-vocal firing rates scaled specifically with the temporal complexity of a COM call, increasing with the number of syllables to be produced. This may reflect the behavioral state and expectancy of the animal. *C. perspicillata* is frequently engaged in vocal interaction with their conspecifics[5,45,46], and emitting a higher number of syllables may increase the likelihood of eliciting a reply, with stronger pre-vocal activity reflecting readiness to process a vocal return signal. Given that the time intervals between individual COM syllables are very short, typically below 10 milliseconds, precise encoding of every syllable as an individual element may not be feasible for the AC, as discussed above, and is more likely performed in brainstem areas with shorter latencies to motor output. Instead, the AC may use rate coding to signal syllable clusters without representing the exact number of elements in a sequence.

Following vocalization onset, neurons exhibited both increased and decreased responses relative to baseline activity, with the population average reflecting an overall excitation in firing. This aligns with observations in other mammals, where AC neurons can show excitation or suppression during vocal production[13–15]. Compared to acoustic playback, neuronal activity during self-generated sounds can often be reduced, as has been shown in rodents and primates[17,47,48]. Although we did not test responses to external sounds, similar suppression mechanisms may also act in the bat AC, likely contributing to the regulation of auditory processing during vocalization.

Taken together, our results demonstrate that single neurons in the bat AC, a highly specialized system for auditory processing, encode information about future and ongoing vocalizations related to their spectral and temporal content. These findings highlight the role of the AC as an active hub in shaping vocal behavior and preparing the network for sensory feedback of self-generated sounds.

## Methods
### Animal details and surgical procedures
The study was conducted on ten awake *Carollia perspicillata* bats (four males, six females). All experimental procedures were in accordance with European regulation for animal experimentation and were approved by the Regierungspräsidium Darmstadt (experimental permit #FU-1126). Bats were obtained from the colony at Goethe University Frankfurt. Animals were housed in a temperature-controlled room maintained at 28 °C with a humidity level of approximately 60% and an inverted 12-hour light-dark cycle. Animals used in experiments were kept physically isolated from the main colony, but with acoustic contact.

Prior to surgical procedures, bats were anaesthetized with a mixture of ketamine (10 mg/kg, Ketavet, Pfizer) and xylazine (38 mg/kg, Rompun, Bayer). A local anesthetic (ropivacaine, 2 mg/ml, Ratiopharm GmbH) was applied subcutaneously on the scalp. A rostro-caudal incision was made in the skin, and skin and muscle tissue were carefully removed to expose the skull. A metal rod ( ~ 1 cm length, 0.2 cm diameter) for later head-fixation was attached to the skull above the frontal midline using UV-glue (Permaplast LH Viscous Flow, M + W Dental). The AC was located by means of well-described landmarks such as prominent blood vessels[25]. A craniotomy ( ~ 1 mm diameter) above the left AC was made by carefully cutting the skull while leaving the dura mater intact. At the end of the surgery, local anesthetic (ropivacaine, 2 mg/ml, Ratiopharm) was applied around the wound. After at least three days of recovery following the surgery, experimental recording sessions took place on a maximum of 4 days per week and lasted no longer than four hours per day. During recording, water was given to the animal every 1–1.5 h.

### Electrophysiological and acoustic recordings
Recordings were performed in an electrically shielded and acoustically isolated chamber. Awake animals were placed on a custom-made holder and head-fixated for the time of the experiment. An ultrasound microphone (model 4135, Brüel & Kjaer) was placed ~10 cm in front of the bat to record vocalizations. Acoustic signals were amplified (Nexus 2690, Brüel & Kjaer) and then fed into a soundcard (ADI-2 Pro, RME) for A/D conversion with 384 kHz sampling rate. Recordings were controlled using the software BatSound Pro (v2.1, Pettersson Elektronik AB).

For neuronal recording, a 16-channel silicon probe (A1x16 with 50 µm spacing between channels, NeuroNexus) was inserted in the right AC perpendicular through the brain surface and slowly lowered using a piezo micromanipulator (PM-101, Science products GmbH, Hofheim, Germany) until the last channel on the probe was no longer visible and the channels spanned across all cortical layers (750 µm). We mainly targeted the high frequency subregions of the *C. perspicillata*'s AC[25]. Reference electrodes (silver wires) were placed underneath the skull above the dura mater ipsilaterally to the silicon probe above the frontal cortex. The probes were connected to a micro-preamplifier (MPA 16, Multichannel Systems, MCS GmbH, Reutlingen, Germany), and acquisition was done with a 32-channel portable system with integrated digitization (10 kHz sampling frequency) and amplification steps (model ME32, Multi Channel Systems MCS GmbH). Acquisition was monitored and stored in a computer using the MC_Rack Software (Multi Channel Systems MCS GmbH, Reutlingen, Germany). Electrophysiological and acoustic data were aligned using TTL pulses sent simultaneously to the sound card and the neuronal recording system.

Neuronal and acoustic data were analyzed using custom-written MATLAB (2023b) scripts (see below).

### Acoustic stimulation
To test the frequency tuning in the auditory cortex, pure tones (10 ms, 0.5 ms rise/fall time) at a frequency from 10 to 90 kHz in steps of 5 kHz were played at 65 dB SPL. These 17 pure tone stimuli were presented 30 times each in a pseudo-random order. The sound was generated with a sound card (ADI-2 Pro, RME) with a sampling rate of 192 kHz, connected to an audio amplifier (model RB-1050, Rotel) and played through a speaker (ScanSpeak Revelator, R2904/7000, Avisoft Bioacoustics) mounted 10 cm away from the bat's right ear (contralateral to the neuronal recording site). Prior to tone production, the speaker was calibrated using a microphone (model 4135, Brüel & Kjaer) that recorded at 16-bit and 384 kHz of sampling frequency with a microphone amplifier (Nexus 2690, Brüel & Kjær). For calibration, the root mean square (RMS) level of each presented pure tone (5 to 95 kHz in steps of 1 kHz) was converted to dB SPL by comparing against the RMS of a 1 kHz reference tone at 94 dB SPL generated with a calibrator device (model 4231, Brüel & Kjaer).

During recording sessions of spontaneous vocalizations, a short play-back series of previously collected distress syllables by *C. perspicillata*[7] was

presented approximately every 10 minutes to keep the bats awake and engaged.

## Analysis of vocal recordings

Acoustic files were first analyzed in Avisoft SAS Lab Pro software (Avisoft Bioacoustics) for detecting vocalizations. The recordings were high pass filtered (cut-off: 5 kHz; eight-order Butterworth filter). Vocalization units were initially detected using an automatic procedure that recognized all threshold crossings (4% of the maximum amplitude of the recording) that lasted longer than 1 ms. Each detected sound was then inspected manually regarding the spectrogram profile and was labeled as echolocation (ECHO) call, communication (COM) call, or noise. While ECHO calls were highly stereotypical with two downward frequency-modulated harmonics and a peak frequency around 80 kHz, COM calls were more variable but typically had a peak frequency lower than 50 kHz. Around every detected vocalization, a period of 1 s (500 ms before, 500 ms after onset) was selected for further inspection. Amplitude and spectrogram profiles of these time periods were checked for any further vocalization or noise that had not passed the initial amplitude threshold detection (meaning the sounds could be of lower amplitude or of shorter duration than the original threshold). Such manually identified sounds were then labeled as ECHO, COM or noise and added to the list of sounds in the dataset. Only vocalizations with a silent period (no call, no noise) of 500 ms before their onset were used for further analysis. Depending on the 500 ms period following each of these calls, they were classified into one of four categories: (1) SIN ECHO–a single ECHO pulse with no following vocalization of any kind for at least 500 ms after onset; (2) SIN COM–a single COM syllable with no following vocalization of any kind for at least 500 ms after onset; (3) SEQ ECHO–a sequence of at least 3 ECHO pulses within 500 ms after onset of the first call; (4) SEQ COM–a sequence of at least 3 COM syllables within 500 ms after onset of the first call. Vocalizations with two syllables or pulses did not fall into any of these categories but were included in the analysis on syllable or pulse quantity (Fig. 3). Note that in this terminology, we use the word "pulse" to describe individual ECHO calls, while "syllable" is used to describe individual segments in a SEQ COM. One syllable always had a clear offset where the amplitude went back to baseline for at least 1 ms before the onset of another vocalization. If no breaks were visible, such a vocalization element was classified as one syllable. With this definition, a SIN COM call has one syllable. Vocalizations were excluded from the analysis if the audio segment contained mixed categories within 500 ms after onset (e.g., two COM syllables and one ECHO pulse).

## Analysis of neuronal recordings

Each recording session was spike sorted with Kilosort 4[49]. Units were refined manually based on visual inspection of spike waveforms as well as auto- and cross-correlograms of each cluster using phy. Across all sessions, we identified 396 neuronal units. Individual bats contributed a comparable number of neurons to the dataset (Fig. S1F).

## Frequency tuning

To assess a neuron's frequency tuning profile, the firing rate of every neuron was calculated in a 50-ms bin following pure tone presentation and averaged for every tone frequency. The bin with the highest firing rate indicated the neuron's best frequency (BF). Based on their frequency tuning profile, neurons were classified into low frequency tuned (LF) neurons, high frequency tuned (HF) neurons and multi-peak (MP) neurons, as previously described[27]. Neurons were LF if the normalized firing rate was higher than or equal to 0.6 at any frequency lower than 50 kHz and lower than 0.6 for all frequencies higher than or equal to 50 kHz. HF neurons were those in which the normalized firing rate was above 0.6 for any frequency higher than or equal to 50 kHz and lower than 0.6 for all frequencies lower than 50 kHz. If the normalized firing rate exceeded 0.6 for any tone in both frequency bands ( < 50 kHz and ≥ 50 kHz), these neurons were classified as MP. To visualize responses to different frequencies, peri-stimulus time histograms (PSTH)

were created from 300 ms before until 300 ms after tone presentation. For every neuron, we also computed the bandwidth of the tuning curve in octaves as

$$BW = \log_2(f2/f1)$$

with f1 and f2 as frequencies around the tuning curve peak in which normalized firing rates dropped to 0.5. For MP neurons, we calculated the bandwidth of both peaks and then summed the two values.

The quality (Q) factor was computed as following:

$$Q = BF/(f2 - f1)$$

## Single neuron analysis to vocalizations

To analyze neuronal activity preceding and following the different vocalization categories (see above), we first identified neurons from recording sessions in which bats emitted at least two different categories of calls. To account for variability across call events, neurons were included only if their activity was recorded during a minimum number of call events per category. This minimum event number was set to 5 for comparing SIN ECHO vs. SIN COM and for comparing SIN COM vs. SEQ COM. As fewer SEQ ECHO were recorded, this criterion was relaxed for comparing SIN ECHO vs. SEQ ECHO and SEQ COM vs. SEQ ECHO to a minimum call event number of 3 per category. For every neuron, we then created PSTHs from 500 ms before until 500 ms after each call with 50 ms bins and averaged firing rates within each call category. These firing rates were then baseline corrected by subtracting firing rates in a 500 ms-period before the first bin (meaning 1 second until 501 ms before each call). For each of these neurons, we also calculated modulation indices, which indicate increased or decreased firing to one call category. To this end, we calculated firing rates within a 500 ms-period directly before call onset (pre) or within a 500 ms-period directly after call onset (post) and averaged these firing rates within a call category. Modulation indices were then computed as following:

$$Modulation\ index_{call\ category} = \frac{firing_{call\,1} - firing_{call\,2}}{firing_{call\,1} + firing_{call\,2}}$$

We also calculated these modulation indices using every 50-ms-time bin of the PSTHs to analyze changes in firing over time.

We calculated an additional modulation index to assess changes in firing rate in the post-vocal window compared to baseline as following:

$$Modulation\ index_{post-vocal} = \frac{firing_{post-vocal} - firing_{baseline}}{firing_{post-vocal} + firing_{baseline}}$$

We further examined the neurons' activity to different calls using principle component (PC) analysis. We first computed activity vectors by z-scoring firing rates across call events and concatenating them for the two respective call categories. This yielded one vector per neuron (time x call category). We then performed PC analysis (Matlab function "pca") across these neurons and extracted the first 3 PC scores. To examine the contribution of specific neurons to the population trajectories, we identified neurons that discriminated the most (high discriminator) or the least (low discriminator) between the two call categories. We compared firing rates across call categories pre and post call onset using the absolute values of the modulation indices described above. The 30% of neurons with the highest absolute modulation indices were classified as high discriminators, whereas the 30% of neurons with the lowest absolute modulation indices were classified as low discriminators. To investigate PC scores with these neuron groups removed, we replaced firing rates of high discriminator or low discriminator neurons in the dataset by their means and then projected the modified datasets into the original PC space applying the following

equation:

$$s = (firing - \mu) \times coef$$

where $\mu$ is the mean firing of each neuron, *coef* contains the PC loadings of the full dataset (output of Matlab function "pca" performed on the original dataset) and *s* are the resulting PC scores.

We then compared the difference of the PC trajectories between call categories for the original dataset, for the dataset with high discriminator neurons removed, and for the dataset with low discriminator neurons removed by computing the RMS Euclidean distance (*d*) of PCs 1-3 across time with the following equation:

$$d = \sqrt{\frac{1}{T}\sum_{t=1}^{T}\sum_{k=1}^{K}(s_{call1,k}(t) - s_{call2,k}(t))^2}$$

where $s_{call1,k}(t)$ and $s_{call2,k}(t)$ are the scores of the two call categories along the *k*-th PC at time bin *t*.

To analyze neuronal activity as a function of emitted syllables or pulses, we first identified call events according to the number of COM syllables or ECHO pulses during a 500-ms window time segment after the onset of the first call. We then identified neurons that were recorded during a minimum number of these call events. For COM calls, we used a criterion of at least 3 different syllable numbers with a minimum of 4 call events each. As fewer SEQ ECHO were recorded, we relaxed this criterion for ECHO calls to at least 3 different pulse numbers with a minimum of 2 call events each. For each call event, we computed the firing rate in a 500 ms window pre and post onset of the first call. These firing rates were then correlated with the number of COM syllables or ECHO pulses for every neuron.

### Population analysis to vocalizations

To further evaluate neuronal activity in relation to the number of syllables or pulses across sessions, we analyzed the firing rates of the whole neuronal population. To this end, we used all recorded call events and computed the spike count of every neuron in 500 ms pre and post windows. Through this method, we could obtain multiple spike counts per call event, depending on the number of neurons that was recorded during each event. In total, we found 3727 instances for COM calls (call events x neurons) and 2207 instances for ECHO calls. All spike counts were then z-scored using the mean spike count and standard deviation across 100 randomly selected 500-ms time segments of the respective neuron. Z-scored spike counts were then fit to the number of syllables or pulses using a linear regression model (Matlab function "fitlm").

To decode call categories from neuronal population activity over time, we used spike counts of each neuron to each call event from 500 ms before until 500 ms after call onset in 25 ms bins convolved with a Gaussian kernel (sigma = 50 ms) and z-scored to the spike rates for all call events of each respective neuron, yielding an activity vector to every instance (call events x neurons). We then trained support vector machine (SVM) binary classification models (Matlab functions "templateSVM" and "fitcecoc" with linear kernel and a 0.1 box constraint) on these activity vectors to decode the call category in pairs of two (e.g., SIN ECHO vs. SIN COM). To train the model and test its performance, we randomly picked 80% of the available activity vectors for each call category and averaged over groups of 15 instances in each category to decrease variability for the model. We then segmented the vectors in bins of 100 ms to decode call category for each bin over time. For each iteration, one activity vector of each category was set aside for later testing, and the model was trained on the remaining vectors. The call categories of the two removed vectors were predicted using the trained model. This was repeated for every vector pair (leave-one-out method) and the percentage of correct decoded call categories was computed after the last iteration. The whole procedure was repeated 20 times, each time picking a new random set of probability vectors. We also generated control models, in which training was performed on the same datasets, but call category labels of each probability vector had been randomized before training.

### Reporting summary

Further information on research design is available in the Nature Portfolio Reporting Summary linked to this article.

## Data availability

The data supporting the findings of this study have been deposited in a GIN repository, which is publicly accessible at https://doi.org/10.12751/g-node.44wmeq[50].

## Code availability

The code used for data analysis and figure generation has been deposited in a GIN repository and is publicly accessible at https://doi.org/10.12751/g-node.44wmeq[50].

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

## Acknowledgements

We thank Gisa Prange and the veterinary team at the Institute for Cell Biology and Neuroscience at Goethe University Frankfurt for their support.

## Author contributions

J.C.H. conceived and supervised the project. D.R. and J.C.H. performed data collection. S.S.B. performed data analysis and wrote the first draft of the manuscript. All authors revised and commented on the manuscript.

## Funding

J.C.H. discloses support for the research of this work from the DFG Heisenberg Program #525004430 and #525183217, from DFG Project #520617944 and #520223571, from DFG Project #532521431, and from DFG project #275755787. Open Access funding enabled and organized by Projekt DEAL.

## Competing interests

The authors declare no competing interests. Julio Hechavarría is a Guest Editor for Communications Biology, but was not involved in the editorial review of, nor the decision to publish this article.
