## [Transparent Peer Review file · Communications Biology]

Neurons in the bat auditory cortex encode class and complexity of future vocalizations

Corresponding Author: Dr Susanne Bahl

Version 0:

Reviewer comments:

Reviewer #1

(Remarks to the Author)

This manuscript by Bahl et al. examines the role of the auditory cortex (AC) in the planning and structuring of vocal behavior in the bat *Carollia perspicillata*. Rather than treating AC as a purely sensory or feedback-processing region, often emphasized in the context of vocalization-related suppression, the authors provide convincing evidence that it is actively engaged during the pre-vocal phase to encode specific features of upcoming calls. This notion is very intriguing, especially in vocal specialists such as bats, and have broader implications to other mammals as well. In particular, using single-unit recordings in awake animals, the authors report that AC neurons encode (i) the class of forthcoming vocalizations (echolocation vs. communication) several hundred milliseconds before vocal onset, (ii) the temporal complexity of communication sequences (syllable count) in the pre-vocal period, and (iii) the number of pulses in echolocation bouts after call onset. These effects are systematically related to neuronal frequency tuning, yielding a coherent picture in which AC carries structured, predictive information about future vocal output. This set of findings is compelling and of high impact. Additionally, I was impressed by how careful the experimental design is. The dataset is also substantial and the analyses are appropriate and convincing. Overall, the study represents a very valuable contribution to our understanding of sensorimotor integration in vocal systems and will be of broad interest to researchers working on auditory processing, vocal communication, and predictive coding. I truly enjoyed reading the manuscript and my suggestions are relatively minor, primarily geared towards placing this work in the broader context and some improvements in clarity for the general audience. A few points could further strengthen the manuscript:

First, the authors should more explicitly distinguish the reported pre-vocal activity from generic corollary discharge or motor-induced suppression effects widely described in rodents and primates. While this distinction is implied, it is central to the novelty of the work: the signals described here are information-bearing and specific to call category and temporal structure, not merely global gain modulation. A clearer statement in the Results or Discussion would help avoid conceptual ambiguity. On that note, I am curious if similar findings would be obtained in other brain areas that have been shown to activate during vocal productions in bats, such as in the frontal cortex during vocal communication in Egyptian fruit bats (Rose et al., *Science*, 2021) and in sensory neurons in the midbrain superior colliculus of big-brown bats (Kothari et al., *eLife*, 2018). While not critical for the publication of this work if the author have recordings from additional areas it would be helpful to assess the specificity of the responses they observe. Again, only if this data exists and if so, I believe it would substantially strengthen the impact of this work.

Second, and in relation to the point above, the Discussion would benefit from a broader framing within predictive and planning-related neural systems. The demonstration that AC encodes future vocal events parallels recent findings that the hippocampus in freely flying bats represents future spatial trajectories during navigation (Dotson & Yartsev, *Science*, 2021). Given that echolocation is tightly coupled to navigation (and indeed it has been shown that auditory objects are represented in the bat hippocampus – see Krishna, ... & Moss, *Current Biology*, 2025), it is plausible that predictive coding in AC and hippocampus reflects coordinated planning of sensory sampling and movement. Even a brief mention of this possibility would place the present results into a richer systems-level context of how predictive coding might be a general feature for animals that rely on rapid sensory information to adjust their high speed maneuvering during flight.

Third, while frontal cortical inputs are discussed as a likely source of preparatory signals, alternative or complementary subcortical pathways (e.g., superior colliculus and periaqueductal gray corollary discharge circuits) are not considered. As noted above, this might still be part of the general circuitry and I think it is worth mentioning. A short acknowledgment of these circuits would provide a more balanced view of the underlying mechanisms. In this context, the head-fixed preparation necessarily limits the range of natural vocal behaviors. The authors may wish to briefly discuss how the reported coding principles might extend to freely moving or flying conditions, particularly during social interactions where call structure and

behavioral context are more complex. Such discussion, while optional as I understand there could be length limitation of the journal format, can be of value for the broader field.

In summary, this is a technically strong and conceptually interesting study that significantly advances the view of sensory cortex as an active component of action planning. With some improvements in conceptual framing, potentially some additional data (if those exist, and if not, pointing to the value of such data for future studies) and discussion of circuit mechanisms and ecological context, the manuscript will make a solid contribution to the literature, and I strongly support its publication.

Michael Yartsev

Reviewer #2

(Remarks to the Author)

In this manuscript, Babl et al. exploit the remarkable vocal abilities of Seba's short-tailed bat to investigate the role of the auditory cortex (AC) in generating vocalization categories. The authors performed extracellular electrophysiological recordings in the AC of ten awake *Carollia perspicillata* during spontaneous vocalizations to determine whether AC activity predicts the type of vocalization about to be produced. The vocalizations fell into two main categories: echolocation (ECHO) calls and communication (COM) calls. The authors show that a large proportion of recorded neurons exhibit category-specific modulation of activity, with firing rates increasing for either ECHO or COM calls even before vocalization onset. Furthermore, when calls were embedded within sequences, pre-call neural activity varied with the number of pulses or syllables in the sequence. Call-specific firing patterns depended on individual neurons' tuning properties.

First, I would like to commend the authors for a clearly written manuscript and well-executed experiments addressing a timely and important question in the field of animal communication: the role of sensory cortical areas in sensorimotor transformations. The animal model used in this study is unique, and the authors provide a strong justification for its use. Importantly, despite the niche nature of this species, the authors have produced a manuscript with broad relevance and value for the field of auditory neuroscience and animal communication.

My comments are largely minor and can be addressed primarily by expanding on points the authors have already raised in their excellent discussion, along with the inclusion of a few additional plots. I begin with three general comments, followed by several more minor suggestions.

1) I appreciate the distinction made between ECHO and COM vocalizations as broad categories; however, as noted in lines 415–416, “the vocal repertoire of *C. perspicillata* is very diverse and depends on the behavioral context.” In addition, ECHO calls are also known to vary in duration and intensity as a function of behavior. Therefore, while the authors provide nice spectrograms to describe these calls, it would be valuable to offer broader behavioral context to aid in their interpretation.

a. How do these types of COM vocalizations compare to the natural vocal repertoire of *C. perspicillata*? What might their functional roles be—e.g., aggression, distress, food solicitation, pup care, etc.? You don't need to engage in broad speculation, but adding a sentence or two about which natural call types these COM calls most closely resemble (e.g., low-frequency calls associated with aggression or distress that carry over longer distances) would help contextualize their potential ethological significance.

b. How stereotyped are the ECHO and COM calls per individual? Is there some sex difference? A supplementary figure that displays the vocal properties per individual and call would be very informative. I was already amazed about how chatty the bats were in the booth, and even more curious about if they were saying the same thing.

2) This leads me to my second comment, which concerns the concept of “modes,” which is excellently discussed in lines 400–414. One could argue that the differences in firing rate between categories correspond to different modes, and, if so, over what timescales these modes operate. In other words, and perhaps more clearly, an additional supplementary figure showing the distribution of call types within an experiment would be very valuable to rule out this interpretation (e.g., large changes in arousal). This could be further strengthened by examining the probability that one call type follows another.

3) This is perhaps my most substantive comment and relates to the reported changes in firing rates. As stated in the Introduction, self-generated sounds (including vocalizations) are generally accompanied by suppression of activity in the auditory cortex—often in a frequency-specific manner—but this effect is typically demonstrated relative to playback of similar sounds. As noted in the Discussion (lines 398–399), such playback controls were not included in the present study. I believe that this needs to be expanded on to contextualize the findings, I suggest some clarifications below:

a. Line 52: “In primates and rodents, neurons in the AC often attenuate their firing rates during self-generated sounds (Rummell, Klee, and Sigurdsson 2016; Eliades and Wang 2008), which may be mediated by inhibitory efference copies originating in the motor system and could aid in distinguishing self-generated from external sounds.” This is true, but the phenomenon appears to hold for self-generated sounds more generally, not just vocalizations (e.g., Audette et al. 2022, showing frequency-specific suppression of self-generated sounds). Please either expand on the similarities and differences between vocalization and other types of self-generated sounds in this context and include additional relevant citations from Schneider and Mooney on this topic (you have several in the Discussion) to better frame this point. I think if you could introduce those notions earlier on then readers would not be as confused when seeing increases in FR in Figure 2A.

b. Did you consider assessing suppression using a modulation index? This is important because, as mentioned in the Discussion: “Following vocalization onset, we found that neurons on average increased their firing rates compared to the pre-vocal period and baseline activity. This aligns with observations in other mammals, where AC neurons can show excitation or suppression during vocal production (Harmon et al. 2024; Eliades and Wang 2005, 2003).” However, this appears to somewhat contradict the general suppression observed during locomotion and other self-generated behaviors. Clarification is necessary on how these results relate to broader suppression phenomena and whether a modulation index analysis might reveal suppression that is not apparent in mean firing rate comparisons.

Below find more minor assorted comments:

- 4) A PCA trajectory would be nice in 3D space would be very nice to illustrate the separation quantified in the text.
- 5) Line 79: "Specific neuron subtypes specializing" if only SU just remove the subtypes.
- 6) Line 99: can you add the n for sessions and animals here?
- 7) The tuning categories used in this study are very broad, and the calls are well separated in frequency content by multiple octaves. Studies from the Moss lab (e.g., Salles 2021; Lawlor 2025), which use vocalization categories that are much closer in frequency content, find different results—albeit in the inferior colliculus and during playback. It would be valuable to discuss those findings as a caveat to the current interpretation of frequency specificity in your results.

Version 1:

Reviewer comments:

Reviewer #1

(Remarks to the Author)

I appreciate and thank the authors for their thoughtful and thorough revision of the manuscript, as well as for their clear and constructive rebuttal. The authors have taken my previous comments seriously and have substantially strengthened the manuscript in several important ways.

I appreciate the authors' effort to more clearly distinguish the reported pre-vocal activity from generic corollary discharge or motor-related suppression signals. The added clarification in the Discussion effectively highlights that the observed activity is information-bearing and specific to call type and temporal structure, which is central to the conceptual advance of this work. This significantly improves the framing and helps avoid potential ambiguity. I also find that the expanded Discussion placing the results within a broader predictive and planning-related framework is very effective. The connection to hippocampal representations of future trajectories and the idea of coordinated sensorimotor prediction across systems provide a richer and more compelling context for the findings. This addition enhances the general relevance of the work beyond the auditory system. Furthermore, the authors have done an excellent job incorporating a more balanced view of potential circuit mechanisms. The inclusion of subcortical pathways, including midbrain structures such as the superior colliculus and periaqueductal gray, provides a more complete picture of the possible sources of pre-vocal modulation and strengthens the overall interpretation. I also appreciate the added discussion regarding the limitations of the head-fixed preparation and how the reported coding principles may extend to more naturalistic conditions. This is an important point for the field, and the authors address it in a thoughtful and appropriately measured way.

Overall, the manuscript is now clearer, better contextualized, and conceptually stronger. The core findings remain compelling: that neurons in auditory cortex encode not only the category but also the temporal structure of upcoming vocalizations, with predictive signals emerging well before vocal onset. Together with the careful experimental design and rigorous analyses, this work provides an important contribution to our understanding of sensorimotor integration and predictive coding in vocal systems.

I have no further major concerns. I strongly support publication of this manuscript in its current form.

Michael Yartsev

Reviewer #2

(Remarks to the Author)

The authors have provided detailed and thoughtful responses to my comments. I have no further comments. I commend them on this excellent contribution.

Response to the reviewers

Reviewer #1 (Remarks to the Author):

This manuscript by Babl et al. examines the role of the auditory cortex (AC) in the planning and structuring of vocal behavior in the bat *Carollia perspicillata*. Rather than treating AC as a purely sensory or feedback-processing region, often emphasized in the context of vocalization-related suppression, the authors provide convincing evidence that it is actively engaged during the pre-vocal phase to encode specific features of upcoming calls. This notion is very intriguing, especially in vocal specialists such as bats, and have broader implications to other mammals as well. In particular, using single-unit recordings in awake animals, the authors report that AC neurons encode (i) the class of forthcoming vocalizations (echolocation vs. communication) several hundred milliseconds before vocal onset, (ii) the temporal complexity of communication sequences (syllable count) in the pre-vocal period, and (iii) the number of pulses in echolocation bouts after call onset. These effects are systematically related to neuronal frequency tuning, yielding a coherent picture in which AC carries structured, predictive information about future vocal output. This set of findings is compelling and of high impact. Additionally, I was impressed by how careful the experimental design is. The dataset is also substantial and the analyses are appropriate and convincing. Overall, the study represents a very valuable contribution to our understanding of sensorimotor integration in vocal systems and will be of broad interest to researchers working on auditory processing, vocal communication, and predictive coding. I truly enjoyed reading the manuscript and my suggestions are relatively minor, primarily geared towards placing this work in the broader context and some improvements in clarity for the general audience.

Authors' response: We thank the reviewer for their positive and constructive comments on the manuscript. Specific comments are addressed individually below.

A few points could further strengthen the manuscript:

- First, the authors should more explicitly distinguish the reported pre-vocal activity from generic corollary discharge or motor-induced suppression effects widely described in rodents and primates. While this distinction is implied, it is central to the novelty of the work: the signals described here are information-bearing and specific to call category and temporal structure, not merely global gain modulation. A clearer statement in the Results or Discussion would help avoid conceptual ambiguity. On that note, I am curious if similar findings would be obtained in other brain areas that have been shown to activate during vocal productions in bats, such as in the frontal cortex during vocal communication in Egyptian fruit bats (Rose et al., *Science*, 2021) and in sensory neurons in the midbrain superior colliculus of big-brown bats (Kothari et al., *eLife*, 2018). While not critical for the publication of this work if the author have recordings from additional areas it would be helpful to assess the specificity of the responses they observe. Again, only if this data exists and if so, I believe it would substantially strengthen the impact of this work.

Authors' response: We thank the reviewer for this insightful comment. We agree that it is important to distinguish the pre-vocal activity described here from more generic motor-

related modulation or corollary discharge signals. To address this, we have added a clarifying statement in the Discussion emphasizing that the observed pre-vocal activity carries specific information about upcoming call type and temporal structure, and therefore cannot be explained by a non-specific motor-related suppression alone:

“Importantly, the pre-vocal firing pattern observed here cannot be explained by a uniform motor-related modulation as it carries information about upcoming call type and temporal structure. This suggests that AC activity reflects specific aspects of planned vocal output rather than general gain modulation of sensory responses.” (line 390-393)

Regarding the involvement of other brain areas, we agree that this is an important question. While we do not include new recordings from additional regions in the present study, previous work from our group has shown that LFP in frontal cortical areas carry call-type-specific information prior to vocalization (Weineck et al. 2020; García-Rosales et al. 2022), and that neuronal activity in the cerebellum also reflects upcoming vocal output (Hariharan et al. 2024). We agree that directly comparing firing rate modulations across these areas, particularly with respect to call type and temporal structure, would be highly informative and interesting. We feel however that adding another area to the present study would change the focus of the article. Furthermore, our previous work, as mentioned above, focused mainly on vocalization-related LFPs and the experimental setup was not explicitly designed for spike recordings, therefore we fear that results regarding stable unit yield might not be exactly comparable. For the current study, we wanted to start exploring spiking activity in a brain area with well described and less complex neuronal firing patterns, namely the auditory cortex. We plan to investigate spiking activity in less acoustically specialized structures, such as the frontal cortex, in future work.

- Second, and in relation to the point above, the Discussion would benefit from a broader framing within predictive and planning-related neural systems. The demonstration that AC encodes future vocal events parallels recent findings that the hippocampus in freely flying bats represents future spatial trajectories during navigation (Dotson & Yartsev, Science, 2021). Given that echolocation is tightly coupled to navigation (and indeed it has been shown that auditory objects are represented in the bat hippocampus – see Krishna,.... & Moss, Current Biology, 2025), it is plausible that predictive coding in AC and hippocampus reflects coordinated planning of sensory sampling and movement. Even a brief mention of this possibility would place the present results into a richer systems-level context of how predictive coding might be a general feature for animals that rely on rapid sensory information to adjust their high speed maneuvering during flight.

Authors' response: We thank the reviewer for this helpful suggestion. We have now included a paragraph in the discussion that puts the observed pre-vocal activity in a broader framework of encoding and predicting audiospatial information:

“As in echolocating species, vocal production is tightly coupled to navigation, the observed pre-vocal firing patterns may be embedded within a broader framework predicting audiospatial information. Recent findings show that hippocampal spatial and goal coding is informed by ECHO calls and encodes future spatial trajectories during flight (Dotson and Yartsev 2021; Ulanovsky and Moss 2011; Krishna et al. 2025). It is possible that predictive coding in the AC and hippocampus reflects coordinated planning of sensory sampling and movement, particularly during echolocation. Within this framework, the pre-vocal activity

observed here may form part of a more general mechanism that supports rapid sensorimotor integration.” (line 429-436)

- Third, while frontal cortical inputs are discussed as a likely source of preparatory signals, alternative or complementary subcortical pathways (e.g., superior colliculus and periaqueductal gray collary discharge circuits) are not considered. As noted above, this might still be part of the general circuitry and I think it is worth mentioning, A short acknowledgment of these circuits would provide a more balanced view of the underlying mechanisms.

Authors' response: We agree with the reviewer that midbrain networks play an important role in predictive coding of vocalizations and it is possible that these regions interact with the AC and contribute to the observed firing patterns. To our knowledge, the PAG does not project directly to the AC, but input could arise via other midbrain areas, such as the superior or inferior colliculus. We have added a section describing the function of several midbrain areas in vocal production and point to possible interaction with the AC:

“Beyond the cortex, subcortical circuits have been described to play a crucial role in controlling vocal output. The periaqueductal gray is known to be critical for vocal production across mammals (Tschida et al. 2019; Nieder and Mooney 2020) and potentially controls different types of vocalizations through distinct neuronal populations (Suga et al. 1973; Valentine, Sinha, and Moss 2002; Fenzl and Schuller 2002). Furthermore, the bat superior colliculus has been shown to encode specific properties of upcoming ECHO pulses (Sinha and Moss 2007; Wohlgemuth, Kothari, and Moss 2018) and play a crucial role in their production (Valentine, Sinha, and Moss 2002; Schuller and Radtke-Schuller 1990). Apart from the midbrain, also neuronal activity in the bat cerebellum contains information about the upcoming vocalization type (Hariharan et al. 2024). These findings raise the possibility that subcortical networks contribute to pre-vocal modulation in the AC, though the underlying circuitry remains to be determined.” (line 401-411).

In this context, the head-fixed preparation necessarily limits the range of natural vocal behaviors. The authors may wish to briefly discuss how the reported coding principles might extend to freely moving or flying conditions, particularly during social interactions where call structure and behavioral context are more complex. Such discussion, while optional as I understand there could be length limitation of the journal format, can be of value for the broader field.

Authors' response: Regarding the vocalizations during head-fixation, we have now expanded the discussion to better relate our findings to more natural behavioral conditions. Specifically, we clarify that the head-fixed preparation limits the vocal repertoire primarily to short COM calls, whereas freely moving or flying bats are expected to produce a broader range of vocalizations, including more frequent ECHO sequences. We further highlight that the temporal structure of ECHO and COM calls may engage AC activity on different timescales, and propose that in more natural settings, where ECHO sequences are more abundant, this could be investigated more directly. The following was added to the manuscript:

“While these short, low-frequency COM calls have been observed in freely moving animals, their ethological role remains unclear. In more natural conditions, particularly during social interaction or flight, the repertoire is expected to be broader, including a wider range of COM types and more frequent SEQ ECHO.

Our data suggest that firing differences between SIN ECHO and SEQ ECHO emerge mostly after the first pulse is emitted, while in contrast changes between SIN COM and SEQ COM were clearly apparent in the pre-vocal window. These observations may indicate that these vocalizations are encoded at different timescales: As inter-call intervals within SEQ ECHO are typically longer than within SEQ COM, AC neurons could encode each ECHO pulse only shortly before its production, whereas rapidly produced COM syllables may be represented more as one sequence-level event. In freely moving or flying animals, where SEQ ECHO are very abundant, this could be investigated more in detail, as the increased number of sequences could reveal clearer temporal dynamics of spike patterns. Overall, we expect that the identified coding principles extend to natural conditions, but within a richer and more dynamic behavioral and acoustic context.” (line 442-455)

In summary, this is a technically strong and conceptually interesting study that significantly advances the view of sensory cortex as an active component of action planning. With some improvements in conceptual framing, potentially some additional data (if those exist, and if not, pointing to the value of such data for future studies) and discussion of circuit mechanisms and ecological context, the manuscript will make a solid contribution to the literature, and I strongly support its publication.

Reviewer #2 (Remarks to the Author):

In this manuscript, Babl et al. exploit the remarkable vocal abilities of Seba's short-tailed bat to investigate the role of the auditory cortex (AC) in generating vocalization categories. The authors performed extracellular electrophysiological recordings in the AC of ten awake *Carollia perspicillata* during spontaneous vocalizations to determine whether AC activity predicts the type of vocalization about to be produced. The vocalizations fell into two main categories: echolocation (ECHO) calls and communication (COM) calls. The authors show that a large proportion of recorded neurons exhibit category-specific modulation of activity, with firing rates increasing for either ECHO or COM calls even before vocalization onset. Furthermore, when calls were embedded within sequences, pre-call neural activity varied with the number of pulses or syllables in the sequence. Call-specific firing patterns depended on individual neurons' tuning properties.

First, I would like to commend the authors for a clearly written manuscript and well-executed experiments addressing a timely and important question in the field of animal communication: the role of sensory cortical areas in sensorimotor transformations. The animal model used in this study is unique, and the authors provide a strong justification for its use. Importantly, despite the niche nature of this species, the authors have produced a manuscript with broad relevance and value for the field of auditory neuroscience and animal

communication.

My comments are largely minor and can be addressed primarily by expanding on points the authors have already raised in their excellent discussion, along with the inclusion of a few additional plots. I begin with three general comments, followed by several more minor suggestions.

Authors' response: We thank the reviewer for their positive and constructive comments on the manuscript. Specific comments are addressed individually below.

1) I appreciate the distinction made between ECHO and COM vocalizations as broad categories; however, as noted in lines 415–416, “the vocal repertoire of *C. perspicillata* is very diverse and depends on the behavioral context.” In addition, ECHO calls are also known to vary in duration and intensity as a function of behavior. Therefore, while the authors provide nice spectrograms to describe these calls, it would be valuable to offer broader behavioral context to aid in their interpretation.

a. How do these types of COM vocalizations compare to the natural vocal repertoire of *C. perspicillata*? What might their functional roles be—e.g., aggression, distress, food solicitation, pup care, etc.? You don't need to engage in broad speculation, but adding a sentence or two about which natural call types these COM calls most closely resemble (e.g., low-frequency calls associated with aggression or distress that carry over longer distances) would help contextualize their potential ethological significance.

Authors' response: We thank the reviewer for this comment. Most COM vocalizations we record during head-fixation resemble those shown in Fig. 1A (right). While we occasionally observe longer vocalizations with harmonic components, these are extremely rare. The behavioral context of these short, low-frequency calls has not been described to date, and they do not clearly match any vocalizations characterized in previous studies on the repertoire of *Carollia perspicillata* (e.g., Knörnschild et al., 2014). However, we also observe similar vocalizations in acoustic recordings of freely moving animals, for example in experiments in which several bats were placed together in a box. Two examples of such low-frequency COM calls intermixed with ECHO calls are shown below.

The ethological significance of these vocalizations remains unclear and is an active topic of discussion within our research group. Their spectral features look quite different from the

distress calls that we have observed in *C. perspicillata*, e.g. when hand-held. While distress calls peak at ~23 kHz and consist of 10 or more syllables (Hechavarría et al., 2016), the COM calls described here peak around 10 kHz and can often appear as a single syllable. They may however carry an aggressive or dominance-related component as harsh, low-frequency vocalizations have often been associated with such motivational states (Morton, 1977). Resolving the behavioral function of these calls will require systematic behavioral observations paired with acoustic recordings. We have added a sentence to the discussion to note that although similar calls occur in freely moving animals, their ethological role remains unknown:

“While these short, low-frequency COM calls have been observed in freely moving animals, their ethological role remains unclear. In more natural conditions, particularly during social interaction or flight, the repertoire is expected to be broader, including a wider range of COM types and more frequent SEQ ECHO.” (line 442-445)

b. How stereotyped are the ECHO and COM calls per individual? Is there some sex difference? A supplementary figure that displays the vocal properties per individual and call would be very informative. I was already amazed about how chatty the bats were in the booth, and even more curious about if they were saying the same thing.

Authors' response: We agree with the reviewer that investigating the vocalizations at the level of individual animals is informative. We have therefore added a supplementary figure depicting, for each individual, the average spectral power, the peak frequency (Fig. S1 A, C), and the number of syllables or pulses for COM and ECHO vocalizations, respectively (Fig. S1 B, D). Within individuals, both COM and ECHO calls were relatively stereotyped in their spectral features, although some variability was observed across vocalizations. While some individuals were highly vocal producing hundreds of calls, other individuals vocalized more sparsely across recording sessions. Overall, male and female bats contributed a comparable number of vocalizations to the dataset (Fig. S1E). We did observe some sex-related differences in acoustic features (Fig. S1G-H). For both COM and ECHO calls, peak frequencies were higher in males, and males produced more consecutive COM syllables than females. A higher number of syllables specifically in distress call sequences produced by males compared to females has also been reported previously (Gonzalez-Palomares et al. 2021).

2) This leads me to my second comment, which concerns the concept of “modes,” which is excellently discussed in lines 400–414. One could argue that the differences in firing rate between categories correspond to different modes, and, if so, over what timescales these modes operate. In other words, and perhaps more clearly, an additional supplementary figure showing the distribution of call types within an experiment would be very valuable to rule out this interpretation (e.g., large changes in arousal). This could be further strengthened by examining the probability that one call type follows another.

Authors' response: We followed the reviewer's advice and generated a supplementary figure to examine if call types occurred in persistent modes. First, we looked at the distribution of COM and ECHO vocalizations across time in individual recording sessions by computing a

modulation index between these two call types for every 10 second bin (Fig. S1I). Overall, we did not observe long, clearly dominant stretches of one call type. The result was similar also for other bin sizes (from 1 s to 2 min). While some sections contained more COM calls and others more ECHO calls, the overall pattern was mixed. Because individual vocalizations are extremely brief (on the order of milliseconds), most of each session consists of silence, making it unlikely to observe extended stretches dominated by a single call type. These analyses suggest that large-scale, persistent behavioral modes do not explain the observed patterns of neural activity.

We also computed the probability that one call type follows another (Fig. S1J). Although some transitions are statistically more frequent than others (e.g. COM to COM vs. COM to ECHO), these likely reflect the overall prevalence of COM vs. ECHO calls rather than persistent modes. In particular, COM syllables often occurred in sequences, whereas ECHO pulses appeared more often as singles in our dataset (see also Fig. 1J). Therefore these probabilities are likely due to sequence structure and relative call occurrence. This analysis, together with the distribution figure, indicates that call sequences are interspersed and do not form extended periods dominated by a single call type.

3) This is perhaps my most substantive comment and relates to the reported changes in firing rates. As stated in the Introduction, self-generated sounds (including vocalizations) are generally accompanied by suppression of activity in the auditory cortex—often in a frequency-specific manner—but this effect is typically demonstrated relative to playback of similar sounds. As noted in the Discussion (lines 398–399), such playback controls were not included in the present study. I believe that this needs to be expanded on to contextualize the findings, I suggest some clarifications below:

a. Line 52: “In primates and rodents, neurons in the AC often attenuate their firing rates during self-generated sounds (Rummell, Klee, and Sigurdsson 2016; Eliades and Wang 2008), which may be mediated by inhibitory efference copies originating in the motor system and could aid in distinguishing self-generated from external sounds.” This is true, but the phenomenon appears to hold for self-generated sounds more generally, not just vocalizations (e.g., Audette et al. 2022, showing frequency-specific suppression of self-generated sounds). Please either expand on the similarities and differences between vocalization and other types of self-generated sounds in this context and include additional relevant citations from Schneider and Mooney on this topic (you have several in the Discussion) to better frame this point. I think if you could introduce those notions earlier on then readers would not be as confused when seeing increases in FR in Figure 2A.

Authors’ response: We thank the reviewer for this helpful suggestion. We have revised the Introduction to disambiguate neuronal activity during vocalizations, other self-generated sounds and the attenuation of these responses when compared to playback of the same sound. We now emphasize that AC activity to vocalization can include both increased and decreased firing and clarify that responses to self-generated sounds are often reduced relative to externally generated sound playback rather than suppressed below baseline. This is the revised section:

“In primates and rodents, neuronal activity in the AC is typically modulated during vocal production. A recent study in mice could show that different populations of neurons either

increase or decrease their firing rate already several hundreds of milliseconds before the onset of an ultrasound vocalization (Harmon et al. 2024). In monkeys, neurons that reduce their activity during vocalizations seem to be more predominant (Eliades and Wang 2003, 2005). Such modulation is not limited to vocalizations but has also been observed during other self-generated, predictable sounds (Audette et al. 2022). Compared to the response to external sounds, the activity to self-produced sounds can often be attenuated (Rummell, Klee, and Sigurdsson 2016), which may be mediated by inhibitory efference copies originating in the motor system (Schneider, Nelson, and Mooney 2014; Schneider and Mooney 2015).” (line 41-52)

b. Did you consider assessing suppression using a modulation index? This is important because, as mentioned in the Discussion: “Following vocalization onset, we found that neurons on average increased their firing rates compared to the pre-vocal period and baseline activity. This aligns with observations in other mammals, where AC neurons can show excitation or suppression during vocal production (Harmon et al. 2024; Eliades and Wang 2005, 2003).” However, this appears to somewhat contradict the general suppression observed during locomotion and other self-generated behaviors. Clarification is necessary on how these results relate to broader suppression phenomena and whether a modulation index analysis might reveal suppression that is not apparent in mean firing rate comparisons.

Authors’ response: We now computed a modulation index relative to a baseline period (500 ms before the pre-vocal window) to assess both suppressed and excitatory responses at the single-neuron level for each call category (Figure S2K). While some neurons showed suppression, the overall distribution was skewed toward positive values, with a positive median, indicating that the population response was overall dominated by increased firing during vocal production. Computing the modulation index using the pre-vocal window as baseline activity yielded similar results. This is consistent with previous reports showing a mixture of suppressed and excitatory responses in AC during vocal behavior. We point out this finding in the results section:

“To assess whether vocalization-related responses showed overall excitation or suppression following vocal onset, we computed a modulation index relative to a baseline period preceding the pre-vocal window. While both suppressed and excitatory responses were observed, the distribution of modulation indices was skewed toward positive values, indicating a predominance of increased firing rates following the onset of vocal production (Figure S 2K).” (line 135-139)

We also adjusted the discussion as following:

“Following vocalization onset, neurons exhibited both increased and decreased responses relative to baseline activity, with the population average reflecting an overall excitation in firing.” (line 470-471)

Below find more minor assorted comments:

4) A PCA trajectory would be nice in 3D space would be very nice to illustrate the separation quantified in the text.

Authors' response: We now show the 3D PC trajectories and the Euclidean distance over time in Figure S 3A-H.

5) Line 79: "Specific neuron subtypes specializing" if only SU just remove the subtypes.

Authors' response: This was replaced with "neuronal populations". (line 74)

6) Line 99: can you add the n for sessions and animals here?

Authors' response: N were added in line 96-97.

7) The tuning categories used in this study are very broad, and the calls are well separated in frequency content by multiple octaves. Studies from the Moss lab (e.g., Salles 2021; Lawlor 2025), which use vocalization categories that are much closer in frequency content, find different results—albeit in the inferior colliculus and during playback. It would be valuable to discuss those findings as a caveat to the current interpretation of frequency specificity in your results.

Authors' response: We have added a brief note in the Discussion acknowledging that ECHO and COM calls differ both in spectral content and behavioral context, and that these factors cannot be fully disentangled in the current dataset:

"It remains however difficult to fully disentangle whether the observed firing differences between ECHO and COM calls reflect mainly acoustic features, call category, or an interaction of both since these call classes differ both in behavioral context and in their spectral content." (line 426-428).